# The Proof-of-Concept of MBA121, a Tacrine–Ferulic Acid Hybrid, for Alzheimer’s Disease Therapy

**DOI:** 10.3390/ijms241512254

**Published:** 2023-07-31

**Authors:** Emelina R. Rodríguez-Ruiz, Raquel Herrero-Labrador, Ana P. Fernández-Fernández, Julia Serrano-Masa, José A. Martínez-Montero, Daniel González-Nieto, Mayuri Hana-Vaish, Mohamed Benchekroun, Lhassane Ismaili, José Marco-Contelles, Ricardo Martínez-Murillo

**Affiliations:** 1Neurovascular Research Group, Instituto Cajal (CSIC), Ave. Doctor Arce 37, 28002 Madrid, Spain; emelinaruth95@gmail.com (E.R.R.-R.); raquel.herrero.labrador@gmail.com (R.H.-L.); ap.fernandez@cajal.csic.es (A.P.F.-F.); jserrano@cajal.csic.es (J.S.-M.); jamm@cajal.csic.es (J.A.M.-M.); 2Experimental Neurology Unit, Center for Biomedical Technology (CTB), Universidad Politécnica de Madrid, Campus de Montegancedo S/N, Pozuelo de Alarcón, 28223 Madrid, Spain; daniel.gonzalez@ctb.upm.es; 3UT Southwestern Medical Center, Department of Neurosurgery, School of Medicine, Baylor College of Medicine, Rice University, Houston, TX 77005, USA; mhg1@rice.edu; 4Laboratoire de Recherches Intégratives en Neurosciences et Psychologie Cognitive de Besançon, Groupe Chimie Médicinale, Université de Franche-Comté, F-25000 Besançon, France; mohamed.benchekroun@outlook.com; 5Laboratory of Medicinal Chemistry, Institute of Organic Chemistry (CSIC), C/Juan de la Cierva, 3, 28006 Madrid, Spain; iqoc21@iqog.csic.es; 6Center for Biomedical Network Research on Rare Diseases (CIBERER), CIBER, ISCIII, 28029 Madrid, Spain

**Keywords:** Alzheimer’s disease, beta amyloid, APP_swe_/PS1_ΔE9_, cerebral cortex, ferulic acid, hippocampus, MBA121, tacrine

## Abstract

Great effort has been devoted to the synthesis of novel multi-target directed tacrine derivatives in the search of new treatments for Alzheimer’s disease (AD). Herein we describe the proof of concept of MBA121, a compound designed as a tacrine–ferulic acid hybrid, and its potential use in the therapy of AD. MBA121 shows good *β*-amyloid (A*β*) anti-aggregation properties, selective inhibition of human butyrylcholinesterase, good neuroprotection against toxic insults, such as A*β*_1–40_, A*β*_1–42_, and H_2_O_2_, and promising ADMET properties that support translational developments. A passive avoidance task in mice with experimentally induced amnesia was carried out, MBA121 being able to significantly decrease scopolamine-induced learning deficits. In addition, MBA121 reduced the A*β* plaque burden in the cerebral cortex and hippocampus in APP_swe_/PS1_ΔE9_ transgenic male mice. Our in vivo results relate its bioavailability with the therapeutic response, demonstrating that MBA121 is a promising agent to treat the cognitive decline and neurodegeneration underlying AD.

## 1. Introduction

Dementia is a brain disease and a common diagnosis in aging populations [1]. More than 55 million people are living with dementia, and this number will almost triple by 2050 as result of the demographic aging phenomenon [2]. From the biological point of view, ageing is associated with neurovascular unit illness [3] and a wide variety of molecular [1] and cellular damages [4], leading to a gradual decline in physical and mental abilities.

Alzheimer’s disease (AD), a multifactorial disorder considered the main cause of dementia in the elderly population, shows a prevalence of more than 35 million people worldwide, with a significant impact on public health. Worldwide, it is estimated that the number of AD new cases will at least double by 2050, with the proportion of AD cases increasing from 7% to 12%. Reaching an epidemic proportion, AD will substantially rise its socioeconomic burden, becoming a major threat to healthcare in our societies [5]. 

Subjacent neurodegeneration in AD progresses with deterioration of memory and other cognitive domains [6] and polyproteinopathies [7]. Whereas research is still needed to uncover the possible causes of AD, the *β*-amyloid (A*β*) hypothesis was first proposed as the leading cause of the neurodegeneration [8,9]. Later, accumulation and aggregation of specific AD disease-related proteins, such as tau, was evidenced [10]. The extracellular senile plaques and intracellular neurofibrillary tangles (NFTs) result from the accumulation and deposition of the A*β* and the aggregation of hyperphosphorylated tau protein, respectively [11]. A*β* progressive load exerts an increasing toxicity in the surrounding neuropil, leading to the development of dystrophic neurites and glial responses over the clinical course of AD, thereby potentially contributing to cognitive decline [12]. Neuronal dystrophic alterations, neuronal cell loss, astrogliosis, and microgliosis are characteristics of the progression of AD [12,13,14]. The severity of dementia is positively related to the number of NFTs [15]. The allele 4 of the *APOE* gene, located on chromosome 19, has been identified as an important risk factor [16,17]. In contrast, familial AD (5–10%) is an early-onset and inherited autosomal dominant [18]. The main mutations causing familial AD have been identified in the genes that code for the amyloid precursor protein (APP) located on chromosome 21, presenilin 1 (PSEN1), located on chromosome 14, and presenilin 2 (PSEN2), located on chromosome 1 [19,20]. The APP is endoproteolytically processed by *β*-secretase and γ-secretase to release amyloid peptides (A*β*_40/42_), which aggregate to form senile plaques. The C-terminal of A*β*_40/42_ is generated by γ-secretase, whose activity depends on presenilins 1 and 2 [21]. Genetic studies point to an increase in the aggregation and generation of the A*β*_42_ peptide in patients with mutations in the APP gene [22]. Statistics indicate that AD may constitute the ‘pandemic of the 21st century’, being a priority for medical research [6,23]. Thus, it is imperative to increase efforts in medicinal chemistry approaches in the search for molecules able to slow or stop the progress of the disease.

From the therapeutic perspective, antagonists of the *N*-methyl-D-aspartic acid receptor, such as memantine, show a limited efficacy by reducing the pathological levels of the neurotransmitter glutamate [24]. Acetylcholine (ACh) plays an important role in memory function and has been implicated in aging-related dementia [25]. Because of the loss of ACh in the brain of patients [26,27], drugs were developed to improve memory by inhibiting acetylcholinesterase (AChE), the enzyme responsible for the hydrolysis of ACh. The advent of acetylcholinesterase inhibitors (AChEI) like donepezil, rivastigmine, galantamine, and tacrine provided only moderate symptomatic benefits.

Tacrine (THA), the first anticholinesterasic drug, was commercialized in 1993 for the treatment of memory loss and intellectual deterioration in patients with AD [28]. THA showed efficacy improving cognitive deficits, but it also exhibited acute liver injury [29] and other side effects [30], resulting in its withdrawal for the treatment of AD.

Because of the inefficiency of the current therapy to limit the progression of AD, other targets have also been considered, such as *β*-secretase, monoamine oxidases (MAO), serotonin, and sigma-1 receptors.

New multi-target-directed ligands (MTDLs) able to hit multiple targets have been developed [31], showing, besides anticholinesterasic activity, A*β* aggregation inhibition properties, metal chelating, and nitric oxide releasing properties, as well as antioxidant capacity [32]. Among the MTDLs, THA, donepezil hybrids [32,33,34], and ASS234, the latter compound endowed with MAO and ChE inhibition properties [35], have been designed and tested, showing lower or no hepatotoxicity [28,36,37,38] and potential to slow the progress of the pathology [39] with no secondary effects [40].

Ferulic acid (FA) is a natural antioxidant, anti-inflammatory, antimicrobial, and hepatoprotective agent [28], with proven ability to decrease A*β* deposits in transgenic mice [28,41].

New THA–ferulic acid (THFA) hybrids (TFAH) showing good antioxidant properties have been identified [28,36,37,38], but among them, (*E*)-3-(hydroxy-3-methoxyphenyl)-*N*-{8[(7-methoxy-1,2,3,4-tetrahydroacridin-9-yl)amino]octyl}-*N*-[2-(naphthalen-2-ylamino)2-oxoethyl]acrylamide (MBA121) was of particular interest (Figure 1). In a recent study [42], we identified MBA121 as a new multipotent TFAH, showing strong anticholinesterasic and antioxidant properties, no hepatotoxicity, good neuroprotection against toxic insults such as A*β*_1–40_, A*β*_1–42_, and H_2_O_2_, as well as good A*β* anti-aggregation activity, a series of properties that support it as a promising agent for the treatment of AD.

In the present study, the effect of MBA121 on cognition was examined in healthy adult C57BL/6J mice in a model of scopolamine-induced cognitive impairment [43]. Furthermore, we examined the effects of short-term MBA121 treatment on plaque load by histochemistry, in a transgenic APP_swe_/PS1_ΔE9_ mouse model of AD [21]. Our findings confirm that MBA121 is effective in in vivo AD models, and it is a promising therapeutic agent to treat AD.

## 2. Results and Discussion

### 2.1. Relief of Scopolamine-Induced Long-Term Memory Deficit in Mice by MBA121

The passive avoidance (PA) paradigm is traditionally used as a quick and easy way to explore short- and long-term memory. APP_swe_/PS1_ΔE9_ mice are indistinguishable from non-transgenic mice on cognitive tasks at six months of age, but by 18 months of age they perform worse than either single transgenics or wild-type animals [44]. In vivo effects of MBA121 on memory were explored on scopolamine-treated mice to model cognitive impairment [43]. This task is based on contextual memory, in which the hippocampus plays an important role, by observing the association between an environmental context that the animal learns and the avoidance of an aversive stimulus (foot shock). The PA test has been extensively used to evaluate learning and memory in rodent models of central nervous system (CNS) disorders and for screenings of novel chemical entities on memory functions [43,45].

MBA121 was administered to C57BL/6J mice treated with scopolamine in a single dose (3.1 mg/kg), intraperitoneally (ip), 90 min before the electric-shock.

Disturbances of memory-acquisition, consolidation, and recall in scopolamine-induced transient amnesia (memory impairment) in mice mimic a characteristic manifestation of dementia [46]. Scopolamine produces amnesic effects in both rodents and humans, so it is widely used to perform pharmacological tests and study the effects of enhancers in cognitive functions [43].

Cognitive deficits induced by scopolamine (1 mg/kg) were strong. As shown in Figure 2, the latency time in the scopolamine group (Scop) was significantly lower (* *p* < 0.05) than the data obtained in the control group (Control) administered with vehicle. As expected, the donepezil group showed a latency time similar to those observed in the control animals (not shown). It is noteworthy to mention the antiamnesic effect of MBA121 (** *p* < 0.01) on the scopolamine treated group (MBA121/Scop). Attenuation of scopolamine-induced amnesia by MBA121 is of particular concern, as no significant differences in the latency time were found between donepezil/Scop and MBA121/Scop, although a slightly tendency to improvement was observed in the MBA121/Scop group compared with control and donepezil/Scop groups. According to the data obtained during the test, animals treated with MBA121 remained in the illuminated compartment longer time than the animals with cognitive deficit induced by scopolamine. As expected, and according to previous studies, donepezil [46] also showed a longer latency time.

All considered, we can conclude that both MBA121 (3.1 mg/kg) and donepezil (1 mg/kg) can induce similar effects in cognition, and that MBA121 exerts cognitive enhancer effects.

### 2.2. Donepezil vs. MBA121: Locomotor Activity

Cholinergic neurotransmission plays a key role in learning and memory. Donepezil and MBA121 boost the effects of ACh, a neurotransmitter that is noticeably depleted in AD patients. The Open Field task (OF) is frequently used in neurobiology for screening the effect on spontaneous locomotor activity of novel drugs delivery [47]. The OF examines locomotor activity in rodents, providing straightforward quantitative measurement of variables such as distance travelled and speed. As shown in Figure 3, the MBA121/Scop mice were comparably more active than the donepezil/Scop group (* *p* < 0.05), as indexed by the distance travelled (Figure 3A) and speed (Figure 3B) during 5 min tests. This correlates with the PA paradigm, in which a tendency of improvement was observed in the MBA121/Scop group compared with the donepezil/Scop group (Figure 2).

### 2.3. Plate Count Aβ: Reduction of the Deposition of Aβ Plaques in MBA121 Treated APP_swe_/PSEN1_ΔE9_ Mice

The hippocampus is a crucial structure in the brain for memory and cognition [48]. The concomitant cellular mechanism in AD is the synaptic dysfunction that begins in vulnerable areas, such as the hippocampus and the neo-cortex, where the accumulation of the A*β* peptide and the hyperphosphorylation of the tau protein occur. Multi-transgenic animals, such as the APP_swe_/PSEN1_ΔE9_ model used in this study, are routinely employed as in vivo models of AD in pre-clinical research, as they exhibit plaques formation, synaptic loss, and behavioral abnormalities [49]. Plaques become evident in cerebral cortex and hippocampus in three-month old animals and continue to increase up to around 12 months of age, with abundant plaques in the hippocampus, cortex, and thalamus by nine months.

Sections from vehicle/treated APPS_we_/PS1_ΔE9_ mice were histochemically stained following the thioflavin-S staining procedure [50] to evaluate the focal accumulation of amyloid plaques. Following MBA121 treatment, 9.5-month-old APP_swe_/PS1_ΔE9_ mice showed a significant plaque count reduction (* *p* < 0.05), both in the hippocampus and in the cerebral cortex (Figure 4, Figure 5, Figure 6 and Figure 7). Both the number and the average plaque size (area occupied by the plaque) were analyzed. In the hippocampus, the reduction involved large (>500 μm^2^) and intermediate (200–500 μm^2^) plaques (* *p* < 0.05), while small plaques (30–200 μm^2^) were unaffected (Figure 6). In contrast, MBA121 treatment yielded a significant reduction in plaque average count (* *p* < 0.05) in the cerebral cortex affecting all plaque sizes (Figure 7).

## 3. Material and Methods

All procedures with animals were carried out in accordance with the European Communities Council Directive (2010/63/ UE) on animal experiments, under a protocol approved by the Animal Welfare Committee of the Cajal Institute (Madrid, Spain) and by the Institutional Animal Ethics Committee of the Spanish National Research Council (CSIC), adhering to the recommendations of the European Council and Spanish Department of Health for Laboratory Animals (R.D. 53/2013). A special effort was made to reduce the number of animals used in the study, and the number of animals assigned to each group was to be kept to a minimum, to achieve adequate significance as previously reported [43].

### 3.1. Behavioural Experiments

Experiments were carried out in 23 male 12 week old C57BL/6J mice (Harlan) weighing 25–30 g. Scopolamine induced amnesia was induced by the potent antimuscarinic drug scopolamine hydrobromide (1 mg/kg, Sigma) administered ip, as previously reported in our laboratory [43]. To explore the therapeutic efficacy of MBA121 as a cognitive enhancer following the PA test, the hybrid drug was administered at a dose of 3.1 mg/kg, ip. As a positive control, the antiamnesic effect of donepezil (Tocris Bioscience, R&D Systems Inc., Minneapolis, MN, USA) at a dose of 1 mg/kg ip, was assessed in the same model of scopolamine transient cognitive amnesia effect. MBA121 and donepezil, were dissolved in saline buffer containing 0.093% dimethyl sulfoxide (DMSO; AppliChem, Darmstadt, Germany) and scopolamine was dissolved in saline. All drug solutions were independently prepared immediately before use. Different batches of mice were used for each experiment. Animals were distributed into four experimental groups (I–IV). Control group I, (*n* = 6), consisted of vehicle treated mice, labelled as ‘’control’’; Experimental group II (*n* = 5), consisted of scopolamine administered mice, 30 min before electric shock, labeled as “Scop” mice; Experimental group III, consisted of co-administered scopolamine plus donepezil (*n* = 6), labeled as “Done/Scop”, in which donepezil and scopolamine were administered 90 min and 30 min before the electric shock, respectively; Experimental group IV (*n* = 6), co-administered with scopolamine plus MBA121 and labeled as “MBA121/Scop”, in which both MBA121 (3.1 mg/kg, ip) and scopolamine were administered 90 min and 30 min before the electric shock, respectively.

The memory task was performed using commercial equipment operated automatically (PA Step Through, Cat No. 7550, Ugo Basile, Comerio, Italia). The PA test is usually used to investigate emergent therapies aimed to study potential cognitive enhancers [43,50]. In brief, the device consists of two compartments (10 × 13 × 15 cm) connected by a sliding door. One compartment is brightly lit (10 W) (compartment Start) and, on the contrary, the other one is dark and equipped with an electrified grid floor (compartment Escape). Rodents tend to prefer dark environments and will immediately enter the darkened compartment. The inter-trial interval was 24 h. The day before the experiment, mice were placed in the experimental room (with sound and light attenuated) for 1 h before experiments. Each animal was then placed gently in the illuminated compartment of the apparatus; after 5 s the guillotine door was opened, and the animal was allowed to enter the dark module. The maximum latency to move from the illuminated camera to the dark compartment was set to 60 s. After 10 s, the mouse was removed from the dark compartment and placed in its cage. The experiment was performed over two consecutive days (d 1, and d 2). On day 1, scopolamine or saline was administered 30 min before the training session and saline-DMSO (0.093%), MBA121, or donepezil were administered 60 min prior to the administration of scopolamine or saline. During the training session, each mouse was individually placed in the illuminated compartment with the sliding door closed. After 30 s, the door was open so that the mouse could move freely to each room. Once the mouse reached the dark compartment, the sliding door closed automatically, and the animal received an electric foot shock (0.5 mA, 1 s). The mice were then placed into their home cage and, 24 h later, in the second day (d 2), the probe session took place. In the second day, each individual was placed again into the illuminated box and, 5 s later, the sliding door opened with the electric foot-shock switched off in the dark room. The latency in seconds taken by a mouse to enter the dark compartment after door opening during the training and the probe session was automatically determined by the computer device. A cut-off time of 5 min was defined [51].

To rule out the possibility that treatment with MBA121, at the dose used in the PA task, could have a deleterious effect on the locomotor activity of the mouse strain, a study was carried out in a group of animals (*n* = 6). We carried out the robust OF task to analyze the effect of MBA121 treatment on locomotion. The OF allows to measure the characteristics of movements, both in the peripheral and central zones of a 40 × 30 × 30 cm polyvinyl chloride box. Evaluation time was 5 min, and movement analysis were performed using a software Ethovision XT© (Noldus, Wageningen, The Netherlands) following a previously described procedure [47]. Parameters such as distance walked and speed of the animals were quantitatively evaluated. The effect of MBA121 vs. donepezil on locomotor activity was measured in a group of 12 week old C57BL/6J mice. MBA121/Scop treated (*n* = 3) and donepezil/Scop treated (*n* = 3) groups were analyzed. Donepezil/Scop and MBA121/Scop administration was carried out as for the PA test, 90 min and 30 min before placing animals in the arena.

### 3.2. MBA121 Treatment in APP_swe_/PS1_ΔE9_ Mice

Studies on the effect of MBA121 on A*β* load were carried out in male APP_swe_/PS1_ΔE9_ mice characterized by an age-dependent progressive deposition of A*β* plaques in the brain parenchyma [52]. At the histopathological level, the efficacy of the MBA121 molecule on the reduction of A*β* plaque burden in vivo was tested using thioflavin-S staining, a gold-standard procedure for the detection of amyloid plaques in the brain [43,50]. Staining was performed on 35 µm-thick coronal sections taken from previously fixed brains. The study was conducted in a group of mice (*n* = 8), injected with either vehicle (*n* = 4, control) or MBA121 (*n* = 4, treated), starting at the age of 4.5 months, for a 5-months duration. Treatment was carried out subcutaneously using ALZET mini-osmotic pumps (ALZET©, model 2004, Cupertino, CA, USA) that were implanted subcutaneously. The treated group was administered daily with MBA121 at a dose of 0.41 mg/kg/day. The control group was administered with vehicle, DMSO (50%) in ethanol (7.5%) and water. The mini-osmotic pumps were replaced every 28 d. The mini-osmotic pumps are widely used for the continuous administration of drugs in the study of neurodegenerative diseases [53,54]. After 5 months of treatment, rodents were sacrificed, and histological studies were performed following the procedures described below. During treatment, body weight was recorded once a week. After removal, the pumps were checked for complete delivering of the drug solution, by aspirating the remaining solution from the pump using a blunt-tipped filling tube and a 1.0 mL syringe. In all cases, pumps were removed after completion of the delivery time, typically following 28 d after implant.

### 3.3. Tioflavin-S Staining

After 5 months of treatment, the mice were sacrificed to proceed to the statistical evaluation of the number of A*β* plaques generated, both in the cerebral cortex and in the hippocampus. The brains were fixed by transcardial perfusion with phosphate buffered saline for 10 s, followed by 100 mL of a solution of paraformaldehyde 4% (*w*/*v*) in 0.1 M phosphate buffer (PB), pH 7.4. Once extracted, the brains were postfixed in the fixation solution for 4 h at room temperature. Subsequently, they were cryopreserved in 30% sucrose in PB 0.1 M, pH 7.4, at 4 °C until use. Coronal sections of 35 μm thickness were then obtained from the fixed and cryopreserved brain using a cryostat (LEICA CM 1950). The deposits of A*β* were visualized by staining with 0.05% thioflavin-S in 50% ethanol [55]. The sections obtained from the experimental animals were selected between the levels of Bregma AP −1.28 mm and −1.64 mm, according to mouse brain stereotaxic atlas [56]. For each treatment group, five sections were processed, visualized, and photographed with a LEICA AF 6500–7000 microscope.

### 3.4. Statistical Analysis

The mice behavior test was analyzed by applying the non-parametric Kruskal–Wallis test, followed by the Dunn’s multiple comparison test to determine which groups were statistically different from each other. Significant differences were considered at * *p* < 0.05 (Figure 2). The images obtained following the thioflavin-S staining were analyzed using the ImageJ software, open source under the General Public License. Prism 5.0 software was used to perform the Mann–Whitney non-parametric test. The quantification of the amyloid plaques exhibited significant differences (* *p* < 0.05) when comparing the control and the treated groups (Figure 5, Figure 6 and Figure 7). The results are expressed as means ± standard errors of the mean, and we considered that the differences are significant at * *p* < 0.05 (Figure 4).

## 4. Conclusions

Despite tremendous efforts aimed at finding an efficient treatment for AD, no definitive cure has been found so far. Anti-cholinesterase hybrids are being created as MTDLs against AD. The synthesis of novel multi-target directed THA-based derivatives, with lower hepatotoxicity in comparison with THA, possessing a neuroprotective profile for the treatment of AD, has been described [57]. FA, a potent antioxidant with multiple beneficial biological actions that improve behavioral impairment and Alzheimer-like pathology in animal models [41], was considered for the design of competent THA-based hybrids [28,38,42,58,59,60]. TFAHs, efficient in scavenging reactive oxygen species and with suppressed hepatotoxicity by THA, were described to reduce the formation of A*β* plaques in vitro [57]. TFAHs were also reported to be effective in behavioral studies in mice, significantly improving scopolamine-induced cognition impairment [61,62]. This paper represents a part of our effort to evaluate in vivo a new family of TFAHs with potential outcome on the aggregation of A*β* [42]. Our new strategy in drug design resulted in a hit compound, namely MBA121, that exhibits significant cognitive improvements and reduction of the plaque load. The TFAH derivative MBA121 easily surpasses the other TFAHs in hepatotoxicity profiling (59.4% cell viability at 1000 μM), providing good neuroprotection against toxic insults such as A*β*_1–40_, A*β*_1–42_, H_2_O_2_, and oligomycin A/rotenone on SH-SY5Y cells, at 1 μM. MBA121 was identified as a very promising hit compound with high BBB permeation capacity, neuroprotective properties, and strong antioxidant capacity, and exhibited non-hepatotoxicity that exceeds other TFAHs [42]. MBA121 has the advantages of high potency, small size, and practicality. Thus, since it displays much less hepatotoxicity than the other hybrids, MBA121 is a good candidate for further investigations. Further in vivo histomorphological studies in liver will be needed to support the in vitro results on hetatotoxicity by MBA121. Based on previous in vitro experiments showing the efficacy of MBA121 in the reduction of A*β* aggregates formation [42], in this work, we carried out further analyses on the in vivo efficacy of MBA121 on the plaque burden on APPS_we_/PS1_ΔE9_ mice bearing the amyloid pathology characteristic of AD [63], and evaluated the anti-amnesic effect of MBA121 in C57BL/6J mice where amnesia is experimentally induced with scopolamine [64,65,66,67,68]. In male mice, MBA121 was able to significantly reduce plaque counts in the hippocampus and cortical areas after a chronic 5-month treatment starting at 4.5 months of age, and to decrease the learning deficits induced by scopolamine in healthy adult mice. Our in vivo results show a correlation of MBA121 bioavailability and the therapeutic response.

Neuritic plaques exert increasing toxicity in the surrounding neuropil associated with dystrophic neurites and glial inflammatory responses over the clinical course of AD, thereby potentially contributing to cognitive decline [69,70]. Thus, reduction of the plaque load would contribute to reduce the cognitive decline. To address this objective, coronal 35 μm-thick sections taken from treated and control animals were processed for thioflavin-S staining. We focused our study on the hippocampus and cerebral cortex, major areas affected by AD [71,72]. Sections from vehicle- and MBA121-treated APPS_we_/PS1_ΔE9_ mice were processed for comparative purposes to check the efficacy of MBA121 in lowering the plaque count.

Further work is currently in progress in our laboratories to conduct a pharmacokinetic and in-depth in vivo toxicological study of MBA121. While we are confident in the safety profile of MBA121, it is important to further investigate its potential toxicological effects. Previous studies have demonstrated that tacrine hybrids, particularly the TFAHs described in the literature, have exhibited reduced hepatotoxicity compared to tacrine [57,58,61]. Therefore, considering the promising results and the cognitive-enhancing properties of MBA121, this is an attractive compound for initiating a pre-clinical study to explore its potential application in the treatment of AD.

## Figures and Tables

**Figure 1 ijms-24-12254-f001:**
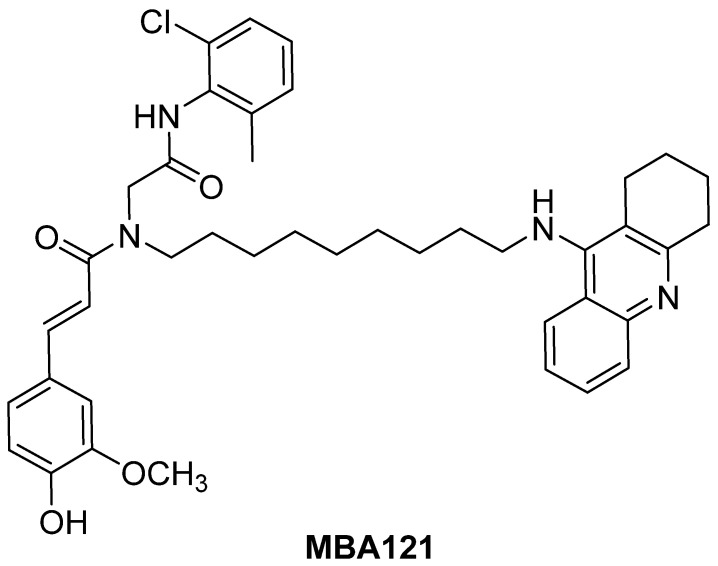
Structure of compound MBA121.

**Figure 2 ijms-24-12254-f002:**
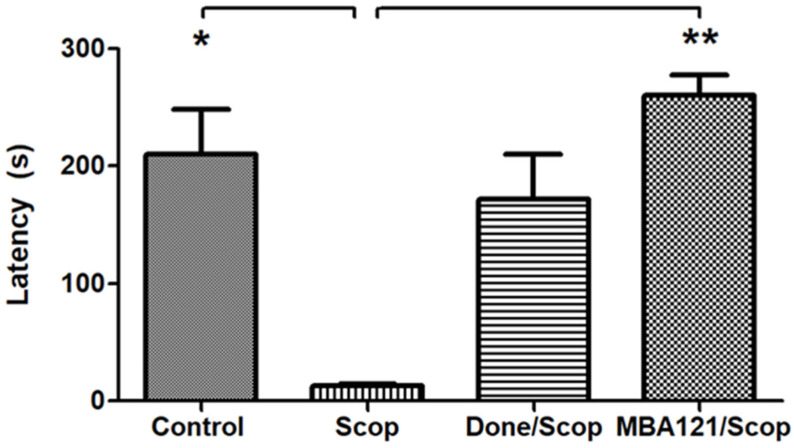
Comparative effects on the latency time of MBA121 and donepezil on the amnesia induced by scopolamine. Bars represent the average of the latency time in seconds (s) corresponding to each experimental group. SEM is represented by error bars. The analysis with the non-parametric test Kruskal–Wallis shows significant differences (* *p* < 0.05) for scopolamine vs control, and (** *p* < 0.01) for scopolamine vs MBA121/scopolamine and for scopolamine vs donepezil+scopolamine. Values are reported as mean ± SEM.

**Figure 3 ijms-24-12254-f003:**
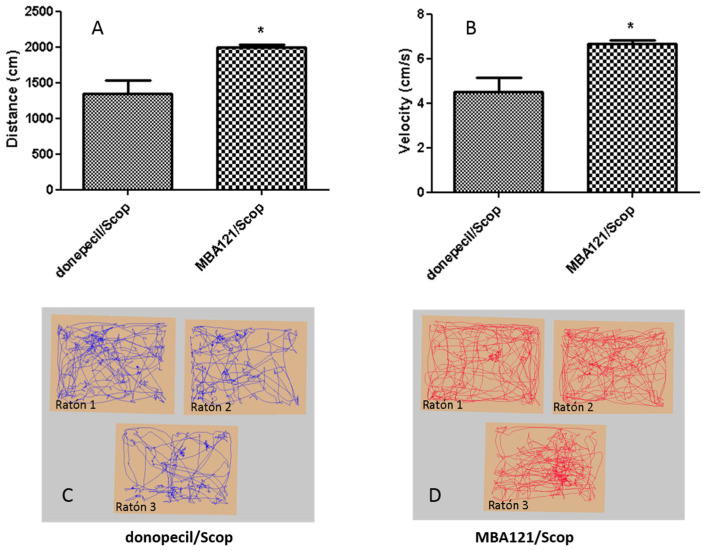
(**A**,**B**) MBA121 administration significantly increased the distance travelled and speed of treated C57Bl/6 J mice compared to that of donepezil-treated mice (* *p* < 0.05). (**C**,**D**) Illustration of the patterns of activity displayed by the donepecil/Scop (**C**) and the MBA121/Scop (**D**) treated groups, measured by using video-tracking. Trace images show animals in both groups entering the center and the periphery of the arena. The (*) indicate statistically significant differences of the distance travelled and speed between both groups. (* *p* < 0.05). Values are expressed as the mean ± SEM.

**Figure 4 ijms-24-12254-f004:**
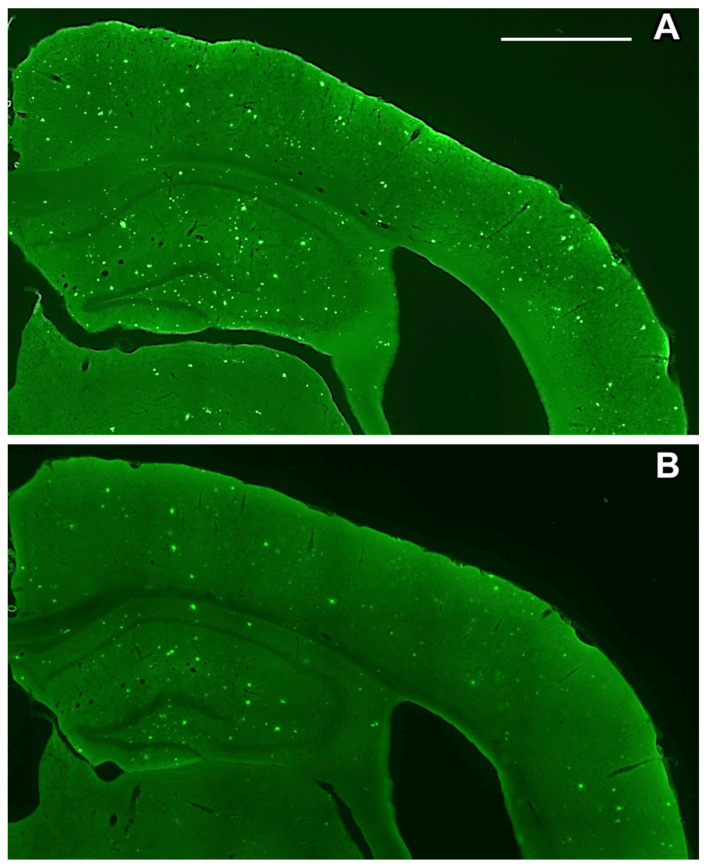
Coronal brain sections of 9.5-month-old APP_swe_/PS1_ΔE9_ mice processed for the detection of A*β* following the thioflavin-S staining. The sections shown are between levels of Bregma −1.28 mm and −1.64 mm. (**A**) Representative image of the distribution of A*β* plaques in the hippocampus and cerebral cortex of a control mice. (**B**) Representative image of the A*β* plaques dissemination in the hippocampus and cortex after five months MBA121 treatment. Scale bar: 900 μm.

**Figure 5 ijms-24-12254-f005:**
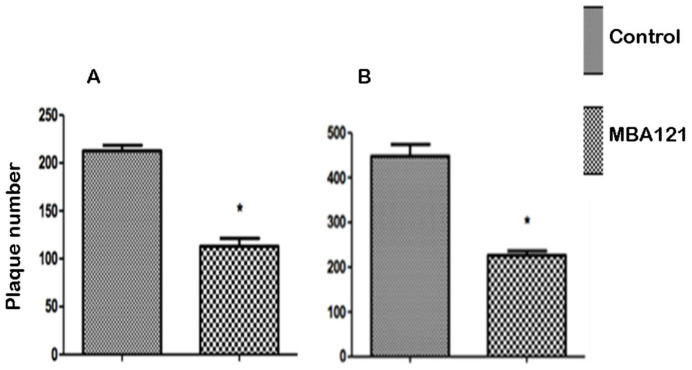
Statistical data regarding the effect of MBA121 treatment on the average of A*β* plaque load in the hippocampus (**A**) and cerebral cortex (**B**) of 9.5-month-old APP_swe_/PS1_ΔE9_ mice treated for 5 months with MBA121. The asterisk (*) indicates statistically significant differences in the hippocampus and cerebral cortex between control and treated groups (* *p* < 0.05). Values are expressed as the mean ± SEM.

**Figure 6 ijms-24-12254-f006:**
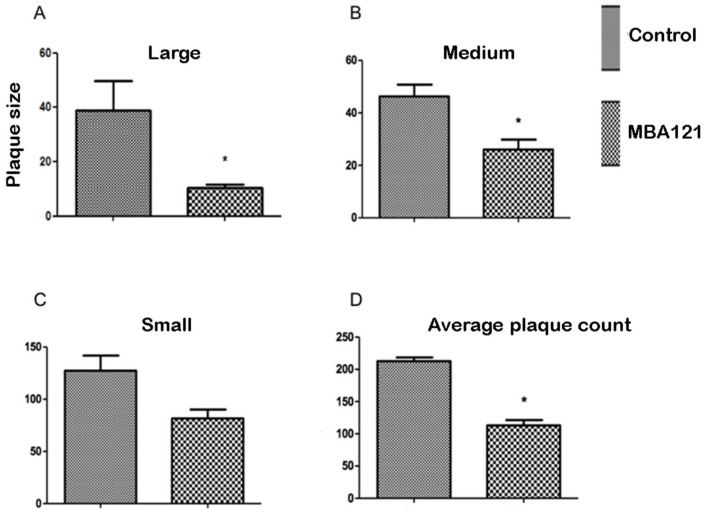
Comparative effects on the A*β* plaque size and count in the hippocampus of 9.5-month-old APP_swe_/PS1_ΔE9_ mice, treated for 5 months with MBA121. The asterisk (*) indicates statistically significant differences between control and treated groups (* *p* < 0.05). The bars represent the average plaque size in μm^2^. Values are expressed as the mean ± SEM. SEM is represented by error bars.

**Figure 7 ijms-24-12254-f007:**
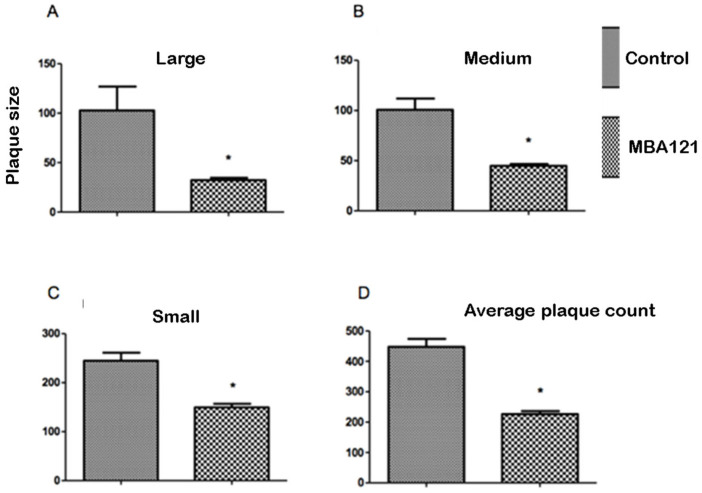
Comparative effects on the A*β* plaque size and count in the cerebral cortex of 9.5-month-old APP_swe_/PS1_ΔE9_ mice, treated for 5 months with MBA121. The asterisk (*) indicates statistically significant differences between control and treated groups (* *p* < 0.05). The bars represent the average plaque size in μm^2^. Values are expressed as the mean ± SEM. SEM is represented by error bars.

## Data Availability

All data are available upon request.

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
