# Peer review of "The Proof-of-Concept of MBA121, a Tacrine–Ferulic Acid Hybrid, for Alzheimer’s Disease Therapy"

_ijms, 2023, doi:10.3390/ijms241512254_

Round 1

Reviewer 1 Report

The paper of Rodiguez-Ruiz and colleagues evaluated the therapeutical potential of a new drug compound in a preclinical mouse model of Alzheimer’s disease. The MBA121 drug is a bivalent compound with anti-cholinesterase activity (Tacrine moiety) and anti-oxidant properties (ferulic acid moiety). When administered to mice presenting scopolamine-induced amnestic symptoms the drug promotes a recovery of (memory) function. When given to APPxPS1 transgenics developing brain amyloidosis, chronic treatment with MBA121 by means of osmotic pumps reduced the number of plaques in mice brains.

The overall content of this (short) report is straightforward, well written but several points have to be considered to envision acceptance of the manuscript.

<MAIN CONCERNS>

-Tacrine was one of the 1st drug developed to treat AD but, due to established hepatotoxicity (and numerous side effects of concern), was withdrawn. Today other anti-cholinesterasic drugs are used (eg Donepezil, Galantamine). The authors claim that their hybrid drug has a good safety potential that may support translational future developments. However no data concerning toxicity (in particular in vivo assessed) are provided, either experimentally or through previous literature reports (cited ref[42] was apparently mostly based on cellular assays). The authors should document this point.

-Tacrine-based hybrids have been engineered and described for years (see PMID: 36675233 for review), including ferulic acid-based compounds. The authors should discuss their results and compare MBA121 properties with existing data from the literature.

-The results obtained in the WT mice under scopolamine are interesting and testify for a pro-memory effect of MBA121. However it is difficult to consider that a pharmacologically induced model of amnesia is an AD model. Did the authors collect behavioral data in the APPxPS1 mice (more robust AD model) under chronic MBA121 regimen?

-The data presented in figure 4 as well as in material & methods are not clear: first this is not a plaque load that is plotted (a load is a surface ratio in %, here the metrics concern number of objects). Second how was plaque number calculated? Raw number/brain region? The right calculation should be density of objects in each region of interest to take into account variability in sampled areas (the same number of objects will give different densities if the surface varies!). The quantitative analysis should be detailed in the relevant section (§3.3), not in the statistics section.

<MINOR POINTS>

-line 22: “going from 12% to 22%” – not understandable ; refer to number of cases (but % of what?)? prevalence (in which population?)?

-line 47: ref [8] cited to illustrate the amyloid cascade hypothesis has no obvious link with this hypothesis.

-line 75-76: “new acetylcholinesterase inhibitors (AChEI) like donepezil, rivastigmine, galantamine and tacrine” – these are drugs with numerous properties except novelty!

-line 84: the sentence should be corrected by something like “Thus, new Multi-Target-Directed Ligands (MTDLs), able to hit multiple targets have been developed [31], and showed, besides anticholinergic activity, …”

-line 99: not “anticholinergic” but “anticholinesterasic”

-Figure 2 should be replaced in the results section, not in the introduction.

-Figure 2: what is the p-value of the difference between scop and scop+donepezil?...

-line 127: “Memory-acquisition, consolidation and recall in scopolamine-induced transient amnesia (memory impairment) in mice is a characteristic manifestation of dementia”. This is overstated and this should be modulated. Amnesia is a constant in AD but is not pathognomonic of this disease as it can have multiple origins. Blockade of cholinergic system is not isomorph to AD. I would remove this sentence or strongly modulate its contents.

-line 137-142: the sentence “Reversed scopolamine-induced deficits…” is not clearly understandable. Rephrase or write several simplified sentences.

-line 173: the sentence on astrocytosis is interesting but has no connections with the study. Remove to avoid an “out of focus” effect.

-line 210: effects of MBA121 treatment on locomotion should be presented (even as a supplementary figure).

-line 213: evidenced instead of evinced?

-line 241: the section title “3.2. Studies on Aβ load” should be changed (eg “MBA121 treatment in APPxPS1 mice”)

-line 242: the 1st sentence has no subject.

-line 247: instead of stating “for 5 months until the age of 9.5 months”, rephrase to something like “initiated at 4.5 months for a 5-months duration”.

-line 255: “After removing, pumps were checked for complete delivering” à explain.

-line 296-300: this paragraph should be summarized and avoid repetitions with material & methods sections. E.g. “MBA121 chronic treatment was able to significantly reduce plaques counts in the hippocampus and cortical areas”.

no specific comments. Some sentences, paragraphs should be rephrased.

Author Response

Point 1: Tacrine was one of the 1st drug developed to treat AD but, due to established hepatotoxicity (and numerous side effects of concern), was withdrawn. Today other anti-cholinesterasic drugs are used (eg Donepezil, Galantamine). The authors claim that their hybrid drug has a good safety potential that may support translational future developments. However no data concerning toxicity (in particular in vivo assessed) are provided, either experimentally or through previous literature reports (cited ref [42] was apparently mostly based on cellular assays). The authors should document this point.

Response 1:  

We thank the reviewer for providing this insightful comment. We acknowledge that the toxicity tests conducted on MBA121 thus far have primarily involved cellular models. In the next stage of our project, we plan to conduct an in-depth toxicity study to further evaluate its safety profile. However, based on the available literature, we are optimistic about the safety potential of MBA121. Several studies have shown that Tacrine hybrids, particularly tacrine-ferulic acid hybrids, have demonstrated reduced hepatotoxicity in vivo compared to Tacrine alone (Chen et al., J. Med. Chem. 2012, 55, 4309-432; Fang et al., J. Med. Chem. 2008, 51, 713-716; Bubley et al., Int. J. Mol. Sci. 2023, 24, 1717). These findings support our confidence in the safety of MBA121.

The following text and references have been added in the Conclusions section (L 385-392):

“Work is currently in progress in our laboratories to conduct a pharmacokinetic and in-depth in vivo toxicological study on MBA121. While we are confident in the safety pro-file of MBA121, it is important to further investigate its potential toxicological effects. Pre-vious studies have demonstrated that tacrine hybrids, particularly the TFAHs described in the literature, have exhibited reduced hepatotoxicity compared to tacrine [57,58,73]. Therefore, considering the promising results and the cognitive-enhancing properties of MBA121, this is an attractive compound for initiating a pre-clinical study to explore its potential application in the treatment of AD.”

Point 2: -Tacrine-based hybrids have been engineered and described for years (see PMID: 36675233 for review), including ferulic acid-based compounds. The authors should discuss their results and compare MBA121 properties with existing data from the literature.

Response 2:

We are now citing reference [57] and compare MBA121 properties with those of a significant number hybrids produced by combination of these two pharmacophores. As a result, it has been described  [42] that MBA121 surpasses the other TFAHs in hepatotoxicity profiling (59.4% cell viability at 1000 μm). The in vivo results of the pharmacological product in this study relates its bioavailability with the therapeutic response, providing support that the multi-targeted molecule MBA121 might be considered as a promising alternative drug of choice to treat the cognitive decline and neurodegeneration underlying AD.

The following text and references have been added in the Conclusions section (L 343-365):

“Despite tremendous efforts aimed at finding an efficient treatment for AD, no definitive cure has been found so far. Anti-cholinesterase hybrids are being created as MTDLs against AD. Synthesis of novel multi-target directed THA-based derivatives, with lower hepatotoxicity in comparison with THA, possessing a neuroprotective profile for the treatment of AD, have been described [57]. FA, a potent antioxidant with multiple beneficial biological actions that improves behavioral impairment and Alzheimer-like pathology in animal models [41], was considered for the design of competent THA-based hybrids [28,38,42,58–60]. TFAHs, efficient in scavenging reactive oxygen species, were described in vitro to reduce the formation of Aβ plaques with suppressed hepatotoxicity by THA [57]. TFAHs were also reported effective in behavioral studies in mice significantly improving scopolamine-induced cognition impairment [61,62]. This paper represents a part of our efforts to in vivo evaluate a new family of TFAHs with potential outcome on the aggregation of Aβ [42]. Our new strategy in drug design have resulted in a hit compound MBA121 that exhibits significant cognitive improvements and reduction of plaque load. TFAH derivative MBA121 easily surpasses the other TFAHs in hepatotoxicity profiling (59.4% cell viability at 1000μM), affording good neuroprotection against toxic insults such as Aβ1-40, Aβ1-42, H2O2, and oligomycin A/rotenone on SH-SY5Y cells, at 1 μm. MBA121 was identified as a very promising hit compound with high BBB permeation capacity, neuroprotective, strong antioxidant capacity and exhibited non-hepatotoxicity that exceeds other TFAHs [42]. MBA121 has the advantages of high potency, small size, and practicality. Thus, since it displays much less hepatotoxicity than the other hybrids, MBA121 is a good candidate for further investigations. Further in vivo histomorphological studies in liver will be needed to support the in vitro results on hetatotoxicity by MBA121”.

Point 3: The data presented in figure 4 as well as in material & methods are not clear: first this is not a plaque load that is plotted (a load is a surface ratio in %, here the metrics concern number of objects). Second how was plaque number calculated? Raw number/brain region? The right calculation should be density of objects in each region of interest to take into account variability in sampled areas (the same number of objects will give different densities if the surface varies!). The quantitative analysis should be detailed in the relevant section (§3.3), not in the statistics section.

Response 3:

For an accurate description of the results, according to the rewiever, we have proceeded to a statistical description of the results in a paragraph that includes the effect of MBA121 in the average of plaque count and size. For this, the following text and references have been added in the resubmitted version (L 200-208):

“Sections from vehicle/treated APPSwe/PS1ΔE9 mice were histochemically stained following the thioflavin-S staining procedure [50] to evaluate the focal accumulation of amyloid plaques. Following MBA121 treatment, 9.5 month-old APPswe/PS1ΔE9 mice disclosed a significant plaque count reduction (*p < 0.05), both in the hippocampus and in the cerebral cortex (Figures 4-7). Both the number and the average plaque size (area occupied by the plaque) were analyzed. In the hippocampus, the reduction involved large (>500 μm2) and intermediate (200-500 μm2) plaques (*p < 0.05), while small plaques (30-200 μm2) were unaffected (figure 6). In contrast, MBA121 treatment yielded a significant reduction in plaque average count (*p< 0.05) in the cerebral cortex affecting all plaque sizes (Figure 7).” 

Added figures 6 and 7 with the data are now shown.

Regarding to minor points:

Point 4: -line 22: “going from 12% to 22%” – not understandable ; refer to number of cases (but % of what?)? prevalence (in which population?)?

Response 4: We have rephrased it as (L 41-44): “Worldwide, an estimated number of AD new cases will at least double by 2050, the proportion of AD cases being thus increased from 7% to 12%, which substantially will rise the socioeconomic burden while reaching an epidemic proportion thus becoming a major threat to healthcare in our societies [5].” 

Point 5: -line 48: ref [8] cited to illustrate the amyloid cascade hypothesis has no obvious link with this hypothesis.

Response 5: In agreement with the referee comment, reference in the resubmitted paper has been changed.

Point 6: line 76: “new acetylcholinesterase inhibitors (AChEI) like donepezil, rivastigmine, galantamine and tacrine” – these are drugs with numerous properties except novelty!

Response 6: In agreement with the referee criticism, the word “new” was omitted.

Point 7: -line 84: the sentence should be corrected by something like “Thus, new Multi-Target-Directed Ligands (MTDLs), able to hit multiple targets have been developed [31], and showed, besides anticholinergic activity, …”

Response 7: Corrected as suggested: lines 85, 86.

Point 8-line 99: not “anticholinergic” but “anticholinesterasic”

Response 8: The term was corrected: line 102.

Point 9: -Figure 2 should be replaced in the results section, not in the introduction.

Response 9: Done

Point 10:-Figure 2: what is the p-value of the difference between scop and scop+donepezil?...

Response 10: (L 139): ** p < 0.01

Point 11: -line 127: “Memory-acquisition, consolidation and recall in scopolamine-induced transient amnesia (memory impairment) in mice is a characteristic manifestation of dementia”. This is overstated and this should be modulated. Amnesia is a constant in AD but is not pathognomonic of this disease as it can have multiple origins. Blockade of cholinergic system is not isomorph to AD. I would remove this sentence or strongly modulate its contents.

Response 11: We have rephrased it accordingly by (L 126-128): “Disturbances in memory-acquisition, consolidation and recall in scopolamine-induced transient amnesia (memory impairment) in mice mimics a characteristic manifestation of dementia {46].”

Point 12: -line 137-142: the sentence “Reversed scopolamine-induced deficits…” is not clearly understandable. Rephrase or write several simplified sentences.

Response 12: Accordingly, we rephrased it as (L 146-152): “Attenuation of scopolamine-induced amnesia by MBA121 is of particular concern as no significant differences in the latency time were found between donepezil/Scop and MBA121/Scop, although a slighttly tendency to improvement was observed in the MBA121/Scop group compared with control and donepezil/Scop groups.”

Point 13: -line 173: the sentence on astrocytosis is interesting but has no connections with the study. Remove to avoid an “out of focus” effect.

Response 13: According with the referee suggestion, the paragraph has been removed.

Point 14: -line 210: effects of MBA121 treatment on locomotion should be presented (even as a supplementary figure).

Response 14: Locomotor activity was measured carrying out an Open Field task (OF) in a group of animals (n=6), a procedure which examines motor deficits providing quantitative measurement of the locomotor activity in rodents. The OF is frequently used in neurobiology for screening novel drugs targets. In this study, the most straightforward measures include as variables distance travelled and speed. Movement’s disorders were not evinced in MBA121/Scop treated animals compared to donepecil/Scop group. In the OF the MBA121/Scop mice were comparably more active than donepecil/Scop group (*p<0.05) as indexed by the distance travelled and speed during the 5min test. In the passive avoidance paradigm, a tendency to improvement was observed in the MBA121/Scop group compared with donepezil/Scop group.

With regard to this, we have included the following sentences in the Results and Discussion (L 169-180) and in the Methods (L 285-297) sections:

(L 169-179)

2.2. Donepezil vs MBA121: Locomotor activity

Cholinergic neurotransmission plays a key role in learning and memory. Donepezil and MBA121 boosts the effects of ACh, a neurotransmitter that is noticeably depleted in AD patients. The Open Field task (OF) is frequently used in neurobiology for screening the effect on spontaneous locomotor activity of novel drugs providing [47]. The OF examines locomotor activity in rodents providing straightforward quantitative measurement of variables, including distance travelled and speed. As shown in Figure 3, the MBA121/Scop mice were comparably more active than donepezil/Scop group (*p < 0.05), as indexed by the distance travelled (Figure 3A) and speed (Figure 3B) during 5 min test. This correlates with the Passive Avoidance paradigm, in which a tendency of improvement was observed in the MBA121/Scop group compared with donepezil/Scop group (Figure 2).”

(L 282-294)

“To rule out the possibility that treatment with MBA121, at the dose used in the Passive Avoidance task, could have a deleterious effect on the locomotor activity on the mouse strain, a study was carried out in a group of animals (n = 6). We proceeded to carry out the robust OF task to analyze the effect of MBA121 treatment on locomotion. The OF allows measuring characteristics of movements, both in the peripheral and central zones of a 40x30x30 cm polyvinyl chloride box. Evaluation time was 5 min, and movement analysis were performed using a software Ethovision XT© following a previously described procedure [47]. Parameters such as distance walked and speed of experimental animals were quantitatively evaluated. The effect of MBA121 vs donepezil on locomotor activity was measured in a group of 12­week­old C57BL/6J mice. MBA121/Scop treated (n = 3) and donepezil/Scop treated (n = 3) groups were analyzed. Drug administration, donepezil/Scop and MBA121/Scop, was carried out as for Passive Avoidance test, 90 min and 30 min before placing animals in the arena."

Point 15 -line 213: evidenced instead of evinced?

Response 15: Done; line 49.

Point 16: the section title “3.2. Studies on Aβ load” should be changed (eg “MBA121 treatment in APPxPS1 mice”).

Response 16:  Accordingly, we introduced the following sentences (L 295-298):

“3.2. MBA121 treatment in APPswe/PS1ΔE9 mice

Studies on the effect of MBA121 on Aβ load were carried out in male APPswe/PS1ΔE9 mice  characterized by a progressive deposition of Aβ plaques in brain parenchyma with age [52]”.

Point 17: -line 242: the 1st sentence has no subject.

Response 17: Corrected; lines 296, 297

Point 18: -line 308: instead of stating “for 5 months until the age of 9.5 months”, rephrase to something like “initiated at 4.5 months for a 5-months duration”.

Response 18:  (L 303), we rephrased the sentence accordingly.

Point 19: -line 255: “After removing, pumps were checked for complete delivering” a explain.

Response 19: We included the paragraph (L 311-315): “After removing, pumps were checked regarding complete delivering of the drug solution. For this, the solution was aspirated from the pump using a blunt-tipped filling tube and a 1.0 ml syringe. In all cases, pumps were removed after completion of the time of the delivery, typically following 28 days after implant”.

Point 20: -line 296-300: this paragraph should be summarized and avoid repetitions with material & methods sections. E.g. “MBA121 chronic treatment was able to significantly reduce plaques counts in the hippocampus and cortical areas”.

Response 20: Repetitions were removed. As for example, in the Conclusions section we introduced the paragraph (L 374-375): “The in vivo results of the pharmacological product related its bioavailability with the therapeutic response.“

Reviewer 2 Report

The article describes the pharmacological efficacy of one of the new conjugates of tacrine and ferulic acid. The combination of these two pharmacophores is found in a significant number of works. The authors mention and cite articles that also investigated such conjugates, but do not compare their results with the results of other studies, do not discuss the benefits of MBA121. And in this case, there are doubts about the necessity and significance of the results presented by the authors.

It seems to me that it would be very useful to supplement the introduction with information about the disadvantages of at least the similar compounds mentioned by the authors, and insert into the discussion a part with a comparison of the properties of MBA121 with other tacrine and ferulic acid conjugates and a clear description of the advantages of MBA121.

In lines 210-213, the authors write about a preliminary assessment of the effect on locomotor functions and the absence of an effect. This is quite important information, so it would be worth describing the method and the results obtained.

The phrase in lines 127-128 (Memory-acquisition, consolidation and recall in scopolamine-induced transient amnesia (memory impairment) in mice is a characteristic manifestation of dementia) seems to me unsuccessful - not  clear - perhaps the first word is missing - disturbances of....?  .

Author Response

Point 1: The article describes the pharmacological efficacy of one of the new conjugates of tacrine and ferulic acid. The combination of these two pharmacophores is found in a significant number of works. The authors mention and cite articles that also investigated such conjugates, but do not compare their results with the results of other studies, do not discuss the benefits of MBA121. And in this case, there are doubts about the necessity and significance of the results presented by the authors.

It seems to me that it would be very useful to supplement the introduction with information about the disadvantages of at least the similar compounds mentioned by the authors, and insert into the discussion a part with a comparison of the properties of MBA121 with other tacrine and ferulic acid conjugates and a clear description of the advantages of MBA121.

Response 1: In the instroduction section we emphasized properties of MBA121 (L 94-105): “Among new THA-ferulic acid (THFA) hybrids (TFAH), showing good antioxidant power [28,36-38), (E)-3-(hydroxy-3-methoxyphenyl)-N-{8[(7-methoxy-1,2,3,4-tetrahydroacridin-9-yl)amino]octyl}-N-[2-(naphthalen-2-ylamino)2-oxoethyl]acrylamide (MBA121) was of particular interest (Figure 1). Ferulic acid (FA) is a natural antioxidant, anti-inflammatory, antimicrobial, and hepatoprotective agent [28], with proven ability to decrease Aβ deposits in transgenic mice [28,41]. Thus, in a recent study [42], we identified MBA121 as a new multipotent TFAH, showing strong anticholinesterasic and antioxidant properties, no hepatotoxicity, good neuroprotection against toxic insults such as Aβ1–40, Aβ1–42, and H2O2, as well as good β-amyloid anti-aggregation compound, a series of properties that support it as a promising agent for the treatment of AD”.

Information has also been addressed in response to Reviwer 1. The new text has been incorporatred in the Conclusions section (L 343-365).

Point 2: In lines 210-213, the authors write about a preliminary assessment of the effect on locomotor functions and the absence of an effect. This is quite important information, so it would be worth describing the method and the results obtained.

Response 2: This information has been addressed in response to Reviwer 1. The new text has been incorporatred in pages (L 169-179) and (L 282-294).

Point 3: The phrase in lines 127-128 (Memory-acquisition, consolidation and recall in scopolamine-induced transient amnesia (memory impairment) in mice is a characteristic manifestation of dementia) seems to me unsuccessful - not clear - perhaps the first word is missing - disturbances of....?  .

Response 3: Included (L 126-128): “Disturbances of memory-acquisition, consolidation and recall in scopolamine-…….”
